# *Hibiscus sabdariffa*, a Treatment for Uncontrolled Hypertension. Pilot Comparative Intervention

**DOI:** 10.3390/plants10051018

**Published:** 2021-05-19

**Authors:** Marwah Al-Anbaki, Anne-Laure Cavin, Renata Campos Nogueira, Jaafar Taslimi, Hayder Ali, Mohammed Najem, Mustafa Shukur Mahmood, Ibrahim Abdullah Khaleel, Abdulqader Saad Mohammed, Hasan Ramadhan Hasan, Laurence Marcourt, Fabien Félix, Nicolas Vinh Tri Low-Der’s, Emerson Ferreira Queiroz, Jean-Luc Wolfender, Marie Watissée, Bertrand Graz

**Affiliations:** 1Antenna Foundation, Avenue de la Grenade 24, 1207 Geneva, Switzerland; alcavin@antenna.ch (A.-L.C.); rnogueira@antenna.ch (R.C.N.); bertrand.graz@unige.ch (B.G.); 2The Iraq Health Access Organization (“IHAO”), District 901, Abu Nawas ST, Baghdad, Iraq; jaafar.taslimi@iraqhao.org (J.T.); hayderajasim@gmail.com (H.A.); mohammednajem61@yahoo.com (M.N.); Mustafa.Shukkur@gmail.com (M.S.M.); ibrahimrangi2@gmail.com (I.A.K.); abdulqaderalhori@gmail.com (A.S.M.); hrh199413@yahoo.com (H.R.H.); 3School of Pharmaceutical Sciences, University of Geneva, CMU, Rue Michel Servet 1, 1211 Geneva 4, Switzerland; Laurence.Marcourt@unige.ch (L.M.); Fabien.Felix@unige.ch (F.F.); Nicolas.Low-Ders@unige.ch (N.V.T.L.-D.); Emerson.Ferreira@unige.ch (E.F.Q.); Jean-Luc.Wolfender@unige.ch (J.-L.W.); 4Institute of Pharmaceutical Sciences of Western Switzerland (ISPSW), University of Geneva, CMU, Rue Michel Servet 1, 1211 Geneva 4, Switzerland; 5WStats Limited, 8, The Mons, Winchester SO23 8GH, UK; mwatissee@wstats.co.uk

**Keywords:** *Hibiscus sabdariffa*, Iraq, hypertension, IDP, refugees

## Abstract

In Iraq, in 2019, there were about 1.4 million Internally Displaced Persons (IDP); medical treatments were often interrupted. The feasibility of using *Hibiscus sabdariffa* (*HS*) decoction to curb hypertension was evaluated. A multicentric comparative pilot intervention for 121 participants with high blood pressure (BP) (≥140/90 mmHg) was conducted. Participants of the intervention group (with or without conventional medication) received *HS* decoction on a dose regimen starting from 10 grams per day. BP was measured five times over six weeks. The major active substances were chemically quantified. Results: After 6 weeks, 61.8% of participants from the intervention group (*n* = 76) reached the target BP < 140/90 mmHg, compared to 6.7% in the control group (*n* = 45). In the intervention group, a mean (±SD) reduction of 23.1 (±11.8) mmHg and 12.0 (±11.2) for systolic and diastolic BP, respectively, was observed, while in the control group the reduction was 4.4 (±10.2)/3.6 (±8.7). The chemical analysis of the starting dose indicated a content of 36 mg of total anthocyanins and 2.13 g of hibiscus acid. The study shows the feasibility of using HS decoction in IDP’s problematic framework, as hibiscus is a safe, local, affordable, and culturally accepted food product.

## 1. Introduction

*Hibiscus sabdariffa* (*HS*) has been used both as a food product and as medicine in different societies and is popularly known as bissap in Africa, karkadé in the Middle East, flor de Jamaica in Central America, and sometime roselle in Europe. Although used for centuries in traditional medicine for hypertension control and as a diuretic [1], the first clinical trial was published only in 1999 [2] in Iran, followed by the work of Herrera-Arellano et al., in Mexico where *HS* calyces are widely used in beverages [3,4]. Since then, other clinical and pre-clinical studies, with a variety of purified extracts and isolated compounds, have testified to its anti-hypertensive effect and investigated molecular mechanisms [5,6]. The anthocyanins-rich fraction (mainly delphinidin-3-*O*-sambubioside and cyanidin-3-*O*-sambubioside) was found to inhibit the angiotensin-converting enzyme (ACE) activity by competing with the active site in a dose-dependent manner [7]. The hibiscus acid-rich fraction exhibits a vasorelaxant effect on rat aorta through an anti-calcic mechanism [8]. Anthocyanins and hibiscus acid appear as the active compounds responsible for the antihypertensive effect of the *HS* calyx. Regarding the traditional use of *HS*, although much progress has been made, questions remain concerning the most affordable pharmaceutical form: Aqueous decoction of *HS* calyces (known as tea/brew). As *HS* brew is known and accessible even for the poorest patients, we dedicated efforts in researching a practical use of *HS* decoction and observed a sustained clinical effectiveness for 6 months when used by hypertensive patients without standard (conventional) medication [9]. During the same study, a similar effect was observed in the three treatment groups: Tablets made of the dried powdered calyx, versus decoction, versus standard treatment. In a pilot study conducted with refugee in Jordan, use of *HS* decoction during 1 month as an adjuvant therapy for patients with uncontrolled hypertension, whether they also used standard medication or not, was associated with promising results: 38% reached the target blood pressure (BP) and 65% of them had systolic blood pressure (SBP) lowered by at least 10 mmHg [10].

In Iraq, by the end of 2019, there were about 1.4 million Internally Displaced Persons (IDP) and 286,900 refugees; medical treatments were often interrupted [11]. In this context, for those with uncontrolled hypertension, *Hibiscus sabdariffa* decoction was seen as a potential solution. *HS* appeared as a clinically validated, locally available treatment against high BP, with a potential to be beneficial in some cases alongside the continued use of conventional treatments, when the latter did not result in the desired BP reduction. In the present work, our objective was to evaluate the effects of recommending, even in such a difficult situation as this study’s setting, *HS* decoction to help control blood pressure.

## 2. Results

### 2.1. Recruitment and Baseline Characteristics

Starting from October 2019, 131 participants were recruited by their medical doctors based on the inclusion criteria. Kirkuk (Shumait center) was randomly assigned to be the control group. The six participants lost to follow up disappeared in the first week. At the time of analysis, four were found in violation of inclusion criteria and were excluded. Thus, 121 participants remained for the final analysis, 45 in the control group, and 76 in the intervention group (35 participants from Abassy-Kirkuk, 32 from Salah al Din, and 9 from Ninewa). Figure 1 shows the participant’s flow.

Demographic and clinical data of participants, as well as BP outcomes after 6 weeks, are presented in Table 1. The two groups differ in gender proportion and participants under standard treatment against hypertension, while mean age is similar between both groups. 

Evolution of SBP over time is presented in Figure 2. These observed differences were also statistically significant after adjustment for blood pressure levels at inclusion (results not shown). After 6 weeks, 61.8% (IC95% 50.0%; 72.8%) of the participants in the intervention group reached the target BP values of < 140 and < 90 mmHg, compared to 6.7% (IC95% 1.4%; 18.3%) in the control group (*p* < 0.00001), whether they were under conventional medication or not. In particular, 64% (30/47) of the participants using conventional treatment reached the targeted BP, while 59% (17/29) of those previously untreated did. In the control group, 5% (2/40) of participants with conventional treatment reached the target BP versus 20% (1/5) of those without medication. Conventional anti-hypertensive medication used by participants (before and during the project) were angiotensin II receptor blockers, calcium channel blockers, ACE inhibitors, beta-blockers, and diuretics, in monotherapy or in association. In the intervention group, 47 participants out of 76 (61.8%) were using conventional medication. The main medications used were monotherapy of calcium channel blockers (23%), ACE inhibitors (23%), or angiotensin II receptor blockers (17%). Associations of two molecules of two different classes of medication were used by 10 participants (21.3%). Differences in the distribution amongst the three “intervention” sites were observed: At one, calcium channel blockers or ACE inhibitors were used in monotherapy in 47% of cases; in a second, angiotensin II receptor blockers or calcium channel blockers in monotherapy represented 65% of uses, and at the third, 37.5% of patients used ACE inhibitors in monotherapy. In the control group, 40 participants out of 45 (88.9%) used conventional medication. In this group, the main treatment, for 37.5% of the participants, was an association of an angiotensin II receptor blockers and a diuretic; 12.5% of the participants used angiotensin II receptor blockers in monotherapy, 12.5% took an ACE inhibitor, 7.5% a beta-blocker and 7.5% a calcium channel blocker. 

In the intervention group, 67% (31/46) of females and 53% (16/30) of males reached the target BP after the 6 weeks. In the control group, none of the 22 females reached the target BP, while 3 males out of 23 did. In the intervention group 89.5% (IC95% 80.3%; 95.3%) of participants had at least a 10 mmHg decrease in SBP, versus 42.2% (IC95% 27.7%; 57.9%) in the control group (*p* < 0.00001). In total, 39.5% (30/76) of the intervention group participants had to increase their *HS* dosage to 15 g after a week and three participants increased the dosage to 20 g directly, which brings the number of participants for whom 10 g of hibiscus was insufficient up to 43.4% (33/76). After 2 weeks of treatment, 50% (15/30) of the participants taking 15 g had to increase to 20 g. Out of those, 13 (87%) were using on-going conventional medication.

After a 6-week follow-up, 47 participants in the intervention group reached the target BP. Out of them, 31.9% (15/47) had had to increase the amount of *HS* before they reached the target BP, while for 68.1% of them (32/47), 10 g of hibiscus per day had been sufficient. In the intervention group 7/76 participants reported side effects (and none in the control group); these were abdominal pain, and for five of them, this inconvenience appeared after increasing the dose to 15 g. These symptoms were all mild to moderate and transient. No possible interaction with ongoing medication was noted, taking *HS* decoction exactly as advised, while 28.9% (22/76) of the participants forgot it a few times. Nobody forgot it more than this.

### 2.2. Chemical Composition of HS Decoction

To know the composition of the *HS* decoction given to patients, a determination of total anthocyanins (compounds known to be present in this plant) was calculated according to the protocol described for the quantitation of anthocyanins in *Vitis vinifera* [12]. A content of 36 mg (0.36%) and 57 mg (0.38%) was determined in the decoction prepared from 10 and 15 g of *HS,* respectively. HPLC-PDA-ELSD and UHPLC-PDA-MS analyses were also carried out to provide a chemical profile of the decoction used (Figure 3).

Using this approach, it was possible to dereplicate the known compounds neochlorogenic acid (1), chlorogenic acid (2), and cryptochlorogenic acid (3) based on their MS and UV spectra [13]. Since these isomeric compounds share the same molecular weight and were not isolated, the elution order could be different. Interestingly, chlorogenic acid was already described for its antihypertensive properties in a biological assay with hypertensive rats [14]. Anthocyanins, which are known to absorb at 520 nm, were detected in a very low amount, which is consistent with the quantitation performed. This result is important since a previous in vitro study attributed the antihypertensive activity of the *HS* to these compounds [7]. The ELSD detection showed a very intense peak assigned to hibiscus acid (4). This compound, described for its vasorelaxant effect on the rat aorta [8], was also quantified by NMR. These analyses showed that this compound is present in a very large quantity in decoction with contents of 2.13 g (21.3%) and 3.23 g (21.5%) from 10 g and 15 g of *HS,* respectively.

## 3. Discussion

In this pilot comparative intervention, we evaluated the effect of *HS* decoction recommended on a dose regimen adapted to response, along with a health awareness lecture, as compared to a control group with health awareness lecture alone. In both groups, participants kept their usual conventional medication if any. At the end of the 6-weeks follow-up, nearly two-thirds of participants using *HS* (61.8%) reached target BP (< 140/90), and a mean reduction of > 20mmHg of SBP and > 10 mmHg of DBP was observed. In the control group, these figures were much lower (6.7% reached the target BP; BP mean reduction 4.4 mmHg for SBP and 3.6 for DBP). Our data suggest that neither the gender of the patient nor an eventual ongoing conventional medication seem to impact on the effect of *HS* decoction on BP reduction. For about two-thirds of the participants, 10 g of hibiscus in 0.5 L per day was sufficient. Most of the participants who needed the highest dose had on-going conventional medication, suggesting that when a treatment is insufficient, it is harder to control the BP. People experiencing forced displacement like refugees and internally displaced persons often face a highly stressful situation. That may in itself be a factor of heightened BP; and disrupted supplies of medication are additional factors of poor hypertension control [15]. This might explain why baseline values of BP were so high (above 150 mmHg) despite the fact that a good part of participants was taking anti-hypertensive conventional medication (albeit with unknown adherence). The slight BP improvement observed in the control group after 6 weeks could be a result of the health awareness lecture (with subsequent reduction of sodium intake, increased physical activity, etc.), or regression to the mean. As all classes of medication were available at each site, the difference of conventional treatments at the various study sites could be explained by the fact that each doctor is free to choose the conventional medication. The effect of *HS* decoction taken simultaneously with each class of antihypertensive agent has not been studied. However, one can assume that *HS*, as a common local beverage, has been often used fortuitously with all of these medications. To our knowledge, no interaction of *HS* decoction with any class of antihypertensive agents has been reported in the literature so far. Most of the intervention group participants (89.5%) had a decrease of at least 10 mmHg in SBP. This has long-term public health implications as the impact of a 10 mmHg (systolic) or 5 mmHg (diastolic) BP decrease can lead to a 22% reduction in coronary heart disease and a 41% reduction in strokes [16]. At the end of the 4 weeks in a pilot intervention among refugees in Jordan, 38% of participants reached the target BP [10]. In the present study, using the same dose regimen and inclusion criteria, but with longer follow-up, larger group size, and inclusion of a control group, we observed a greater magnitude of BP decrease. This could be explained by better teaching of how to use *HS* and improved teaching material such as a take-home video. In a randomized controlled trial in Senegal, a similar mean reduction of 21.8 (± 11.1) mmHg of SBP was observed despite some differences in the protocol, including absence of concomitant conventional medication, longer follow-up (6 months), and a maximal dosage of 15 g (versus 20 g in the present study). However, in the Senegal study, only 45.5% of participants reached the target BP [9]. This might be explained by the different BP baseline: While in the present study, participants started with a mean BP of 152/94 mmHg, in the Senegalese study, participants entered the program with a higher mean BP (158/98). Another possibility could be the beneficial effect of the Health awareness lecture in Iraq. In Iraq, *HS* was generally well-accepted and well tolerated. Since adverse reactions disappeared as soon as participants stopped consuming the decoction, we hypothesized that it is related to its high acidity, due in particular to the high amount of hibiscus acid detected in our phytochemical analysis, as well as ascorbic acid 8.3 mg/10 g [17]. The safety of *HS* decoction and extracts are reported in the literature through different clinical and experimental studies [1], as well as the fact that it is a widely consumed food product. Concerning the therapeutic dose of *HS* in the management of hypertension, no wide consensus has been reached so far. Different doses have been used in various protocols, Aand the lack of chemical analysis of the doses tested constitutes an additional difficulty when deciding how to use this plant in BP management. When the dosage is as low as 2.5 g of *HS* decoction daily, no side effects are reported but the pharmacological effect is also minimum with a mean reduction of 7.43 mmHg for SBP and 6.70 mmHg for DBP [18]. This might be sufficient in some cases. Thirty-two participants out of 76 in our intervention group reached the target blood pressure with the initial dosage of 10 g *HS* decoction/daily; a lower starting dosage could be considered in the future. The 36 mg of anthocyanins determined in 10 g of *HS* can partly explain the antihypertensive effect in participants. Even presented in very small amounts, it was showed that the anthocyanin-rich fraction of *HS* (mainly delphinidin-3-*O*-sambubioside and cyanidin-3-*O*-sambubioside) inhibit the angiotensin-converting enzyme (ACE) activity by competing with the active site in a dose-dependent manner [7]. The large amount of 2130 mg of hibiscus acid (4) in 10 g of *HS* might also help explain the hypotensive effect, as the vasorelaxant activity of pure hibiscus acid via blockade of voltage-dependent Ca2^+^ channels has been shown [8]. Finally, the presence of chlorogenic acid (**2**), even in trace amounts, is also remarkable since this compound demonstrated a reduction of blood pressure in spontaneously hypertensive rats in a dose-dependent manner when administered orally [14]. This is the first time to our knowledge that a phytochemical profile of the *HS* decoction and a quantitation of the assumed active ingredients have been determined and related to its hypotensive effect in the participants of a pilot intervention. The present results reinforce the body of evidence on the role of *HS* against uncontrolled hypertension, whether the patient is using conventional medication or not. It also shows its suitability even in difficult situations such as refugee or IDP settlements.

### Limitations of the Study

Because of the difficulties related to the situation in Iraq, the differences in sphygmomanometer brands used in the different sites, the impossibility of verifying the accuracy of each device, as well as the difference in reading precision depending on the sites (5 or 10 mmHg), certainly had an impact on the accuracy of absolute BP measurements. However, as the same device with the same reading precision was used throughout the study for each patient, the measurement of the relative BP reduction remained consistent. The open-label design and the fact that the control group was not equally distributed to all study sites may have created bias as participants were aware of the treatment allocated to them and there was no placebo as replacement for *HS* in the control group. Although SBP baseline was comparable between both groups, a higher percentage in the control group of patients using conventional anti-hypertensive medication suggests a higher rate of uncontrolled (or difficult to control) hypertension in this group. Also, gender differences between groups and how they adhered to health awareness lectures might have amplified the magnitude of the observed effects. Therefore, results should be interpreted with caution. 

## 4. Materials and Methods

### 4.1. Study Design

This multicentric pilot comparative intervention was conducted in four health centers located in three regions in the center and north of Iraq (Salah al-Din, Ninewa, and Kirkuk) in cooperation with the local NGO Iraq Health Access Organization (IHAO), (Figure 4). One site was chosen randomly to be the control group, while the other 3 sites were merged to be the intervention group. A formal verbal informed consent from each and all the participants were collected during the project as it is done whenever one makes a proposal in the frame of a medical consultation, and no special personal information (aside from information collected for routine medical work) was collected.

### 4.2. Participants

All participants with uncontrolled hypertension registered in the participating health centers were encouraged to join the pilot intervention if following criteria were met:1.Inclusion criteria:Age > 18 years.Systolic BP (SBP) ≥ 140 mmHg and/or diastolic BP (DBP) ≥ 90 mmHg, with or without ongoing antihypertensive medication.No evidence of cardiovascular, renal, or retinal complication.

2.Exclusion criteria:Hypertensive crisis requiring urgent medication.Overt kidney failure (serum creatinine > 1.4 mg/dL).Pregnant or lactating women (excluded on principle, although there is no evidence of any problems encountered with the tested food product).Previous adverse reaction to HS.

### 4.3. Intervention

A health awareness lecture on hypertension management (recommending reduction of sodium intake, physical activity, etc.) was offered to the participants in both intervention and control groups at the beginning of the study. The intervention group received, in addition to the lecture, bags of *HS* calyces and information on how to prepare the decoction. Participants in both intervention and control groups continued to use their regular treatments if any.

*HS* decoction preparation and dosage adjustment scheme: The daily starting dose of decoction for all participants in the intervention group was 10 g of *HS* calyces. In order to create experimental conditions as close as possible to a real potential future context, the dosage was measured by the patient with 5 tablespoons, which together provide approximately 10 grams of *HS*. The patient was instructed on how to prepare the decoction: 10 g poured in 0.5 L of water and boil for about 15 min and advised to drink it throughout the day. After the first week (second medical visit), all participants who did not reach BP <140/90 mmHg were told to increase the daily dose to 15 g, boiled in 1 L (same boiling time). After 2 weeks of *HS* consumption (third visit), if BP was still too high, the recommended dose was increased to 20 g of *HS* boiled for 15 min in 1 L. Participants who missed a follow-up appointment were called to arrange another meeting the same day. If the participant interrupted the treatment for more than 15 days, he/she was excluded from the study and re-enrolled as a new participant, if possible.

### 4.4. Measurement Procedures

All participants were assessed at baseline, and after 1, 2, 3, and 6 weeks. They were questioned about treatment adherence, side effects, and previous or current use of other drugs. BP was measured in accordance with international guidelines [19]. We advised participants to rest for at least 5 min before measuring their BP, which was done with aneroid sphygmomanometer manual arterial pressure measurement devices on both arms. The devices used in the 4 sites were made either by MDF^®^ or Ramstad^®^ and both were graduated with 10 mmHg intervals. Because of the security problems in the country (moving around was often dangerous), the accuracy of devices could not be verified. However, the same measurement device and technique was used throughout the study at each site to ensure consistency of the measurements. In all cases, mean value between both arms was kept for the analysis; the measurement was repeated up to three times per arm in case of discordance.

### 4.5. Ethical Issues

This project was designed as a pilot intervention and was conducted within the framework of the Iraq Health Access Organization (IHAO). Participants were informed about the pilot’s protocol, and oral informed consent was obtained from all participants involved. Because this pilot focused on a locally well-known food product, no ethical committee clearance was required. In addition, measurements were restricted to those of usual clinical care. At the end of the pilot, the results would be presented to all participants and, if found beneficial, *HS* would be recommended to all participants who might benefit from its use.

### 4.6. Outcome Measurements

Primary Outcomes:1.SBP and DBP change after 6 weeks.2.Proportion of participants reaching target BP (< 140/90 mmHg) after 6 weeks.3.Percentage of participants for whom the SBP change was clinically significant (defined as a decrease of at least 10 mmHg).

Secondary Outcomes:4.Adverse events (any new symptoms, plausibility of a causal link).5.Interaction with other medication, plausibility of a causal link.6.Need to increase HS dosage during follow-up.

### 4.7. Statistical Analysis

In the study conducted in Jordan [10], we found that with *HS* decoction, 38% of participants reached target BP after 1 month. In the control group, one might expect 10%. Based on these data and hypothesis, to have a study power of 80% and a two-sided α error of 0.05, with a ratio (*HS*/control) of 3/1, the estimated sample size was 66 and 22, thus a total of 88 participants. With subsequent corrections to take into account potential attrition (20%) and cluster effect (expected 20% sample size increase because of the cluster design, from conservative estimates of cluster effect), the total required sample at inclusion was 127. Data were first recorded on paper and then communicated by internet to the main investigator who transferred them to Excel tables. The statistical analysis was done with Epi Info7 and STATA. The BP reductions from baseline were compared with an analysis of variance (ANOVA), and further analyzed post-hoc taking into account the baseline BP levels. Chi-square and Fisher exact tests were used to compare proportions of clinically significant effects and proportions of participants reaching target BP values.

### 4.8. General Experimental Procedures for the Chemical Analysis

UV spectra were measured on a HACH UV-VIS DR/4000 instrument (Loveland, CO, USA). A Bruker Advance Neo 600 MHz NMR spectrometer equipped with a QCI 5 mm Cryoprobe and a Sample Jet automated sample changer (Bruker BioSpin, Rheinstetten, Germany) was employed for NMR analysis. Chemical shifts are reported in parts per million (δ) using the residual acetone-d6 signal (δH 2.05) as internal standards for 1H NMR; coupling constants (J) are given in Hz. For NMR quantitation the spectra were recorded in a deuterated phosphate buffer (pH 7) containing 0.9 mM of TSP (sodium trimethylsilyl propionate). The decoctions were controlled on a multidetector UHPLC-PDA-ELSD-MS (Waters, Milford, MA, USA) platform fit with a single quadrupole detector (QDa) using heated electrospray ionization. Analytical HPLC was carried out on an HP 1260 Agilent system equipped with a photodiode array (PDA) detector (Agilent Technologies, Santa Clara, CA, USA).

### 4.9. Plant Material

We bought *Hibiscus sabdariffa* L. (Malvaceae) (*HS*) calyces in a reputed herbal shop in Baghdad, Iraq, and ensured its proper identification using a voucher herbarium specimen deposited at the Botanical Garden in Geneva, Switzerland, under the number G00422529. The verification was carried out by the resident botanist based on macroscopic observations. Nigerian *HS* was chosen in Baghdad because it is among the most affordable options in the market and material of the same origin was used in the previous study in Jordan [10].

### 4.10. HS Decoction Chemical Content Analysis

To analyze the chemical content of the two main decoctions used in this study, the samples were prepared according to the *HS* decoction preparation mode described above. The first decoction was made with 10 g of *HS* boiled in 500 mL water for 20 min and the second with 15 g of *HS* boiled in 1000 ml. The samples were then filtered, frozen, and lyophilized affording the following yields: Decoction with 10.00 g yield 4.08 g (40.8%) of the dried decoction, while 7.65 g (51%) was obtained from the *HS* 15 g decoction.

#### 4.10.1. HPLC-PDA-ELSD Analysis of the HS Decoction

The metabolite profiling of the decoction was established by HPLC coupled to photodiode array (PDA) and light scattering (ELSD) detectors. During the decoction preparation with 10 g of *HS*, samples of 20 mL were taken each 5 min. The samples were filtered, frozen, and lyophilized affording the mass of 0.13 g (5 min), 0.11 g (10 min), 0.21 g (15 min), and 0.16 g (10 min). The samples at the concentration of 10 mg/mL (injection of 20 µL) were analyzed with a X Bridge column (250 × 4.6 mm, ID, 5 µm), gradient: 5% MeOH (0.1% formic acid) to 100% MeOH (0.1% formic acid) within 60 min. Flow 1 mL/min. Detection was performed by UV at 254, 320 and 540 nm. The ELSD was set at 45 °C, with a gain of 9. The PDA data were acquired in the range from 190 to 500 nm, with a resolution of 1.2 nm. Sampling rate was set at 20 points/sec.

#### 4.10.2. Quantitation of Anthocyanins

The total anthocyanins content, expressed in 3-glucosidescyanidol, was calculated according to the protocol described for the quantitation of anthocyanins in *Vitis vinifera* [13]. The *HS* decoction prepared with 10 g possess 0.36% of anthocyanins, while the decoction prepared with 15 g possess 0.38%.

#### 4.10.3. Identification and Quantitation of Hibiscus Acid by 1H-NMR

A sample of 3 mg of dry *HS* prepared with 10 g was dissolved in 600 uL of acetone-d6 and analyzed by NMR. The ^1^H-NMR spectrum showed one major compound identified as hibiscus acid by comparison of the NMR chemical shifts with literature [20]. The quantitation of hibiscus acid was performed by ^1^H-NMR [21,22]. To avoid variation of proton chemical shifts, the spectra were recorded in a deuterated phosphate buffer (pH 7) containing 0.9 mM of TSP (sodium trimethylsilyl propionate). This latter was used as an internal reference standard for chemical shift calibration and for quantitation of the active compound in the extract. The signals at δ_H_ 5.17 was used for the quantitation of hibiscus acid. The *HS* prepared with 10 g possess 21.3% of hibiscus acid, while the tea prepared with 15 g possess 21.5% of hibiscus acid.

## 5. Conclusions

For the first time, to the best of our knowledge, a pilot intervention with the anti-hypertensive plant *HS* was carried out using a chemically well-characterized decoction. The chemical study of *HS* decoction demonstrated the presence of traces of chlorogenic acid (2), anthocyanins, and a very large quantity of hibiscus acid (4). According to the literature, all of these compounds might potentially participate in the antihypertensive activity observed in this clinical study [7,8,14]. Because of the presence of hibiscus acid (4) in higher amounts compared to other bioactive compounds, future clinical investigations of this plant should consider this compound as a chemical marker, and especially its quantitation for the development of a quality control analytical method. *HS* decoction appears as effective for the management of uncontrolled hypertension in participants using conventional medication or not, with doses ranging from 10 to 20 g daily, adapted according to clinical response. A lower starting dosage, e.g., 5 or even 2.5 g daily, could be considered in the future, as it might be sufficient for a fair proportion of users. This therapeutic regimen might represent an affordable complementary or alternative anti-hypertensive where *HS* is commonly found. *HS* might be proposed to any hypertensive patient, as adjuvant or alone, for example if the patient wants a natural product or wishes to avoid undesired effects from a conventional treatment. *HS* also appears suitable for use in low-income countries and for those experiencing forced displacement like refugees and IDP.

## Figures and Tables

**Figure 1 plants-10-01018-f001:**
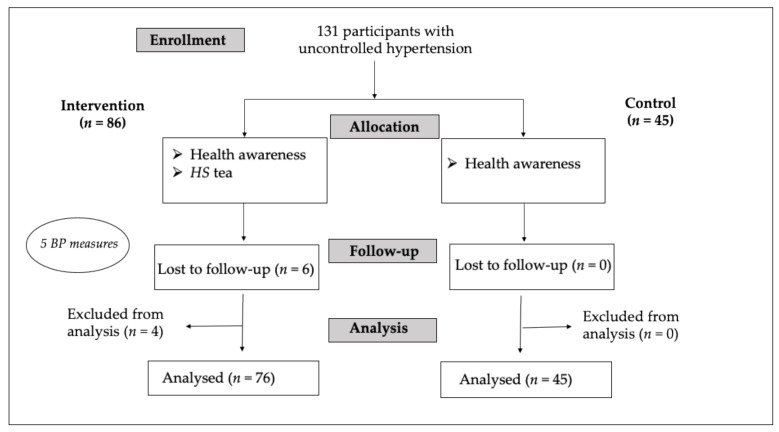
Participant’s flow.

**Figure 2 plants-10-01018-f002:**
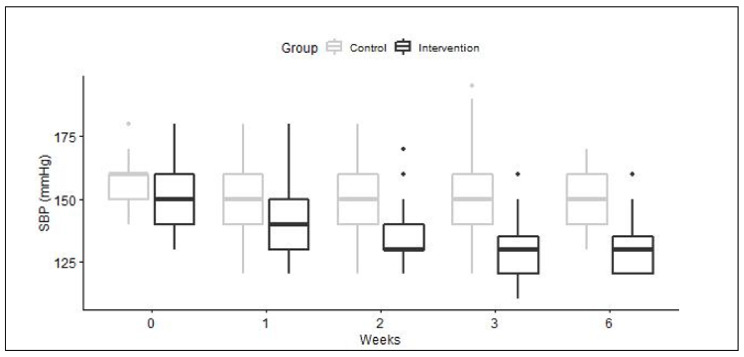
Boxplots of Systolic BP overtime. Median values are black lines in the middle of the box, interquartile range (IQR) is the height of the box; minimum and maximum values (1.5 × IQR) are represented by the lines extending out of the box. The outliers are pinpoints.

**Figure 3 plants-10-01018-f003:**
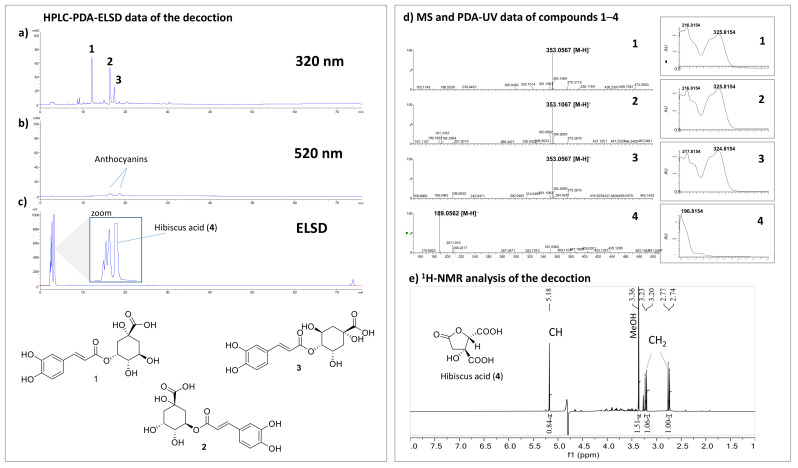
HPLC-PDA-ELSD analysis of *HS* decoction: (**a**) PDA detection at 320 nm; (**b**) PDA detection at 520 nm; (**c**) ELSD detection. UHPLC-PDA-MS analysis (**d**); MS and UV spectrum of compounds 1–3; (**e**) 1H-NMR analysis of *HS* decoction in a deuterated phosphate buffer (pH 7) containing 0.9 mM of TSP (sodium trimethylsilyl propionate). The signals of hibiscus acid (4) are highlighted.

**Figure 4 plants-10-01018-f004:**
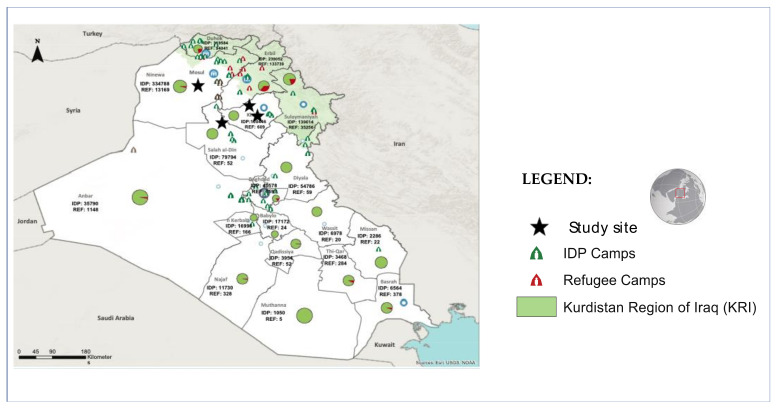
Refugee IDP settlements in Iraq and study sites. Source: Adapted from [11]. IDP—internally displaced persons. January 2020.

**Table 1 plants-10-01018-t001:** Sociodemographic and clinical data.

	Intervention *n* = 76	Control *n* = 45
Age (mean ± SD)	51.0 ± 10.3	53.5 ± 12.8
Gender (%Female)	60.5	48.9
% on anti-hypertensive medication	61.8	88.9
Baseline SBP	151.6 ± 11.7	155.9 ± 10.6
Baseline DBP	93.9 ± 8.8	88.7 ± 12.2
SBP after 6 weeks	128.6 ± 9.2	151.4 ± 10.7
DBP after 6 weeks	81.9 ± 7.7	85.1 ± 7.9
Mean reduction SBP *	23.1 ± 11.8	4.4 ± 10.2
Mean reduction DBP **	12.0 ± 11.2	3.6 ± 8.7

SBP: systolic blood pressure expressed in mmHg (mean ± SD), DBP: diastolic blood pressure expressed in mmHg (mean ± SD); * *p* < 0.00001 ** *p* = 0.00003.

## Data Availability

The data presented in this study are available on request from the corresponding author.

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
