# Peer review of "Hibiscus sabdariffa, a Treatment for Uncontrolled Hypertension. Pilot Comparative Intervention"

_plants, 2021, doi:10.3390/plants10051018_

Round 1

Reviewer 1 Report

  1. Line 22: the full name of BP should be addressed first time shown in the text.
  2. Table 1: The statistics analysis (significant difference?) on listed items between intervention and control groups should be shown. In addition, what does “**” mean?
  3. Figure 3: HPLC-PDA-ELSD analysis is for individual compounds identification and quantitation. A total amount analysis of anthocyanin (36mg anthocyanin in 10g HS?? lines 188-189), phenolics and flavonoid in HS should be conducted and discussed in the manuscript.
  4. Since there are various phytochemicals in HS, how did the authors recognize/prove hibiscus acid in their HS sample is the major active component in this study? Are there other possible anti-hypertension active ingredients in their HS?
  5. Lines 358-359: The literature/reference should be cited.

Author Response

Dear Reviewer,

We are grateful that you have read and commented on our article- Your comments were very useful to improve the article. You will find below the questions raised with answers

I remain at your disposal for any additional information 

Sincerely yours,

Marwah

Reviewer comments 1

  1. Line 22: the full name of BP should be addressed first time shown in the text.

Thanks for noticing it. The description “blood pressure (BP)” was added.

  1. Table 1: The statistics analysis (significant difference?) on listed items between intervention and control groups should be shown. In addition, what does “**” mean?

Thanks for noticing. When the table was transposed this information was lost. Now it is corrected: **p=0,00003

  1. Figure 3: HPLC-PDA-ELSD analysis is for individual compounds identification and quantitation. A total amount analysis of anthocyanin (36mg anthocyanin in 10g HS?? lines 188-189), phenolics and flavonoid in HS should be conducted and discussed in the manuscript.

The HPLC-PDA-ELSD analysis is not only used to identify and quantify organic compounds. It is also a powerful methodology in phytochemistry to obtain an overview of the chemical content of the crude plant extract (chemical profile). The total amount of anthocyanin was determined in the aqueous decoction of Hibiscus sabdariffa (HS). The text was revised concerning this point for clarity. Thank you very much for the comment concerning the phenolics. Indeed, the phenolic derivative chlorogenic acid identified by HPLC-PDA-MS (in trace amounts) in our study has been already described for its antihypertension properties in an in vivo assay with hypertensive rats. A discussion concerning this point was added in the text. Concerning the flavonoids, only traces of anthocyanins were detected. Nevertheless, a discussion concerning this class of compounds was also added in the text.

  1. Since there are various phytochemicals in HS, how did the authors recognize/prove hibiscus acid in their HS sample is the major active component in this study? Are there other possible anti-hypertension active ingredients in their HS?

The hypothesis that hibiscus acid could be one of the “possible active compounds” for the anti-hypertensive effect is based on the article published by Zheoat et al*. In this study, bioguided fractionation from Hibiscus sabdariffa methanolic extract was performed and the authors demonstrate that hibiscus acid has a vasorelaxant effect on the rat aorta. Among the other compounds, present in this polar extract, only hibiscus acid presents this activity. Furthermore, according to our HPLC-PDA-ELSD and 1H-NMR analysis, we demonstrated that this compound is present in unusual high amounts in the water decoction used in the clinical trial. For these different reasons; biological activity combined with presence in high amounts, that we have proposed the hypothesis that this compound could be the “possible active compound” and explain the activity of the decoction. However, the extract also contains traces of chlorogenic acid, a compound already described for its antihypertension properties, and also anthocyanin’s where some compounds also demonstrate antihypertension activities. We have revised the text regarding this point to make it clearer.

*Zheoat, A. M. et al. 2019. Hibiscus acid from Hibiscus sabdariffa (Malvaceae) has a vasorelaxant effect on the rat aorta. Fitoterapia, 134, 5-13.

  1. Lines 358-359: The literature/reference should be cited.

The reference was cited. Thanks

Reviewer 2 Report

Based on previous work the authors initiated a more detailed pilot study to test the feasibility of Hibiscus sabdariffa decoction to treat hypertension. They initiated an open-label, non-randomized intervention trial with 121 participants with uncontrolled hypertension. After six weeks follow-up, they found a reduction of systolic and diastolic blood pressure in the intervention group compared to the control group.

Overall, the results are interesting as they demonstrate the feasibility of a traditional, cheap and easily available natural compound to treat hypertension. This is important in the context that all around the world an increasing number of people exist suffering from diseases like hypertension.

Although, the study has several limitations I would recommend publication after revision.

Major:

The chapter “Limitations of the study” has to be extended, regardless of whether the points have been discussed elsewhere:

  • The control group was not equally distributed to all study sites.
  • There are substantial differences between intervention group and control group: the gender differences may be important; at the beginning, the percentage of patients with conventional anti-hypertensive medication was higher in the control group but the baseline SBP was comparable between both groups suggesting a much higher rate of uncontrolled hypertension in the control group compared to the intervention group.

Minor:

  • The Name of the senior author is not complete.
  • In table 1, the second significance level (**) is not explained.
  • Figure 2: the statistical parameters of the boxplot should be mentioned in the legend.
  • Lines 147-149: what kind of conventional medication has generally been used? Are there differences between intervention group and control group?
  • Methods lines 240-251: How were the patients instructed to take the HS decoction? Equally distributed over the day, or at certain hours? Or once daily? The last option seems not practical for 1 L of decoction.

Author Response

Dear Reviewer,

We are grateful that you have read and commented on our article- Your comments were very useful to improve the article. You will find below the questions raised with answers

I remain at your disposal for any additional information 

Sincerely yours,

Marwah

Based on previous work the authors initiated a more detailed pilot study to test the feasibility of Hibiscus sabdariffa decoction to treat hypertension. They initiated an open-label, non-randomized intervention trial with 121 participants with uncontrolled hypertension. After six weeks follow-up, they found a reduction of systolic and diastolic blood pressure in the intervention group compared to the control group.

Overall, the results are interesting as they demonstrate the feasibility of a traditional, cheap and easily available natural compound to treat hypertension. This is important in the context that all around the world an increasing number of people exist suffering from diseases like hypertension.

Although, the study has several limitations I would recommend publication after revision.

Major:

The chapter “Limitations of the study” has to be extended, regardless of whether the points have been discussed elsewhere:

  • The control group was not equally distributed to all study sites.
  • There are substantial differences between intervention group and control group: the gender differences may be important; at the beginning, the percentage of patients with conventional anti-hypertensive medication was higher in the control group but the baseline SBP was comparable between both groups suggesting a much higher rate of uncontrolled hypertension in the control group compared to the intervention group.

Thanks for this observation. We reformulated the topic “Limitations of the study”.

Limitations of the study

Because of the difficulties related to the situation in Iraq, the difference in sphygmomanometer brands used in the 4 different sites, the fact that the accuracy of each device could not be verified, as well as the difference in reading precision depending on the sites (5 or 10 mmHg), had certainly an impact on the accuracy of absolute BP measurements. However, as for each patient the same device was used throughout the study and with the same reading precision, the relative BP reduction has probably remained consistent. The open-label design and the fact that the control group was not equally distributed to all study sites may have created bias as participants were aware of the treatment allocated to them and there was no placebo as replacement of HS in the control group. This might have had an impact on behaviour and reporting. Although SBP baseline was comparable between both groups, a higher percentage of patients in the control group under conventional anti-hypertensive medication suggests that those patients had a higher rate of uncontrolled hypertension than the ones in the intervention group. Also, gender differences between groups and how they adhere to health awareness might have amplified the magnitude of the effect observed and results should be interpreted with caution.

Minor:

  • The Name of the senior author is not complete.

Thanks for noticing it. His name was corrected.

  • In table 1, the second significance level (**) is not explained.

Thanks for noticing. When the table was transposed this information was lost. Now it is corrected: **p=0,00003

  • Figure 2: the statistical parameters of the boxplot should be mentioned in the legend.

Thanks for this reminder. In order to clarify the statistical parameters used to our readers we included the following description below figure 2.

Median values are black lines in the middle of the box, interquartile range (IQR) id the height of the box; minimum and maximum values (1.5×IQR) are represented by the lines extending out of the box. The outliers are pinpoints.

  • Lines 147-149: what kind of conventional medication has generally been used? Are there differences between intervention group and control group ?

Thanks for this observation. We included more information regarding the conventional anti-hypertensive medication used in both groups in the results section and we also included a paragraph in the discussion about these differences.

Results

Conventional anti-hypertensive medication used by participants (before and during the project) were: angiotensin II receptor blockers, calcium channel blockers, ACE inhibitors, beta-blockers and diuretics, in monotherapy or in association. In the intervention group, 47 participants out of 76 (61.8%) were using conventional medication. The main medications used were monotherapy of calcium channel blockers (23%), ACE inhibitors (23%) or angiotensin II receptor blockers (17%). Associations of 2 molecules of 2 different classes of medication were used by 10 participants (21.3%). Differences in the distribution amongst the 3 “intervention” sites were observed: at one, calcium channel blockers or ACE inhibitors were used in monotherapy in 47% of cases; in a second, angiotensin II receptor blockers or calcium channel blockers in monotherapy represented 65% of uses, and at the third, 37.5% of patients used ACE inhibitors in monotherapy. In the control group, 40 participants out of 45 (88.9%) used conventional medication. In this group, the main treatment, for 37.5% of the participants, was an association of an angiotensin II receptor blockers and a diuretic; 12.5% of the participants used angiotensin II receptor blockers in monotherapy, 12.5% took an ACE inhibitor, 7.5% a beta-blocker and 7.5% a calcium channel blocker.

Discussion

As all classes of medication were available at each site, the difference of conventional treatments at the various study sites could be explained by the fact that each doctor is free to choose   the conventional medication. The effect of HS decoction taken simultaneously with each class of antihypertensive agent has not been studied. However, one can assume that HS, as a common local beverage, has been often used fortuitously with all of these medications. To our knowledge, no interaction of HS decoction with any class of antihypertensive agents has been reported in the literature so far.

  • Methods lines 240-251: How were the patients instructed to take the HS decoction? Equally distributed over the day, or at certain hours? Or once daily? The last option seems not practical for 1 L of decoction.

Thanks for this remark. This important detail was missing. In order to clarify it to our readers we reformulated the phrase (line 285-277) as follow: “The patient was instructed on how to prepare the decoction: 10 g poured in 0.5 L of water and boil for about 15 min and advised to drink it throughout the day.”

Reviewer 3 Report

Were there any measures to determine whether included participants were adherent to medication and lifestyle modification to determine whether they were truly uncontrolled?

The control group seemed to have more participants on drug treatment than the intervention group (89% vs 62%). Does this point to lesser control of blood pressure by lifestyle medication approaches in the control group compared to the HS intervention group? Would this have biased the outcome in favor of the intervention from the onset?

Were control and treated patients matched for age and sex? There seems to be more females in the intervention group. 

Were there any determination of plasma levels of active ingredients or surrogate markers in the intervention group? This may have been useful to determine adherence.

Were there any objective measures to determine adherence to HS in the intervention group?

Line 70 – How were participants recruited?

Line 71 – How were centers randomly assigned to intervention and control groups?

Line 265 – Did all participants had a confirmed diagnosis of hypertension?

Line 311 – Ethical approval is required in ALL interventional studies in humans.

Line 314 – Is it ethical to recommend use of the product from the results of a small, non-blinded, non-randomized and non-controlled, pilot study?

Author Response

Dear Reviewers,

We are grateful that you have read and commented on our article- Your comments were very useful to improve the article. You will find below the questions raised with answers

I remain at your disposal for any additional information 

Sincerely yours,

Marwah

  1. Were there any measures to determine whether included participants were adherent to medication and lifestyle modification to determine whether they were truly uncontrolled?

Thanks for this remark. Most of the participants before the project were in a regular weekly visit to the health care centers, medication adherence, and lifestyle modification were controlled by medical doctors at each center by questioning the patient. Besides those measures, nothing else was made.

  1. The control group seemed to have more participants on drug treatment than the intervention group (89% vs 62%). Does this point to lesser control of blood pressure by lifestyle medication approaches in the control group compared to the HS intervention group? Would this have biased the outcome in favor of the intervention from the onset?

Thanks for this observation. We reformulated the topic “Limitations of the study”.

Limitations of the study

Because of the difficulties related to the situation in Iraq, the difference in sphygmomanometer brands used in the 4 different sites, the fact that the accuracy of each device could not be verified, as well as the difference in reading precision depending on the sites (5 or 10 mmHg), had certainly an impact on the accuracy of absolute BP measurements. However, as for each patient the same device was used throughout the study and with the same reading precision, the relative BP reduction has probably remained consistent. The open-label design and the fact that the control group was not equally distributed to all study sites may have created bias as participants were aware of the treatment allocated to them and there was no placebo as replacement of HS in the control group. This might have had an impact on behavior and reporting. Although SBP baseline was comparable between both groups, a higher percentage of patients in the control group under conventional anti-hypertensive medication suggests that those patients had a higher rate of uncontrolled hypertension than the ones in the intervention group. Also, gender differences between groups and how they adhere to health awareness might have amplified the magnitude of the effect observed and results should be interpreted with caution.

  1. Were control and treated patients matched for age and sex? There seems to be more females in the intervention group. 

Thanks for this observation. We reformulated the topic “Limitations of the study”.

Limitations of the study

Because of the difficulties related to the situation in Iraq, the difference in sphygmomanometer brands used in the 4 different sites, the fact that the accuracy of each device could not be verified, as well as the difference in reading precision depending on the sites (5 or 10 mmHg), had certainly an impact on the accuracy of absolute BP measurements. However, as for each patient the same device was used throughout the study, and with the same reading precision, the relative BP reduction has probably remained consistent. The open-label design and the fact that the control group was not equally distributed to all study sites may have created bias as participants were aware of the treatment allocated to them and there was no placebo as replacement of HS in the control group. This might have had an impact on behavior and reporting. Although SBP baseline was comparable between both groups, a higher percentage of patients in the control group under conventional anti-hypertensive medication suggests that those patients had a higher rate of uncontrolled hypertension than the ones in the intervention group. Also, gender differences between groups and how they adhere to health awareness might have amplified the magnitude of the effect observed and results should be interpreted with caution.

  1. Were there any determination of plasma levels of active ingredients or surrogate markers in the intervention group? This may have been useful to determine adherence.

Thanks for this remark. This has not been done but it’s a nice idea for future studies.

  1. Were there any objective measures to determine adherence to HS in the intervention group?

Thanks for this remark. Most of the participants before the project were in a regular weekly visit to the health care centers, medication adherence, and lifestyle modification was controlled by medical doctors at each center by questioning the patient. Besides those measures, nothing else was made.

  1. Line 70 – How were participants recruited?

Thanks for this remark. We included in the phrase: “Starting from October 2019, 131 participants were recruited.” The sentence: “by their medical doctors based on the inclusion criteria”.

Line 71 – How were centers randomly assigned to intervention and control groups?

Thanks for this remark. Because of the post-war situation in Iraq, the randomization process was based on the frequent and evenly spread road traffic interruptions, resulting in one of the doctors not being able to attend the day when hibiscus treatment was introduced. We decided, then, to assign the center where this doctor belonged to be the control group.

Line 265 – Did all participants had a confirmed diagnosis of hypertension?

Yes, and most of them were taking their classical medication. This was described in our section of inclusion criteria: “All participants with uncontrolled hypertension registered in the participating health centres were encouraged to join the pilot intervention.”

Line 311 – Ethical approval is required in ALL interventional studies in humans. 

Thanks for your comments regarding the Ethical approval. We provided a letter signed for each one of the medical doctors in charge of the pilot evaluation with the requested clarifications, and we added the sentence in the material and method section:  ”A formal verbal informed consent from each and all the participants were collected during the project as it is done whenever one makes a proposal in the frame of a medical consultation, and no special personal information (aside from information collected for routine medical work) was collected.

Line 314 – Is it ethical to recommend the use of the product from the results of a small, non-blinded, non-randomized and non-controlled, pilot study?

Hibiscus sabdariffa is a well-known tea consumed for centuries. In the scientific literature, we can find 2 meta-analyses of trials conducted worldwide (ref). The aim of our study was to evaluate the feasibility of using this product in the difficult context of internally displaced persons in Iraq. We found that when patients have access to a local product that is also affordable and clinically validated, they will be able to adhere to a chronic treatment like hypertension with less economic burden.

 Serban, C., Sahebkar, A., Ursoniu, S., Andrica, F. and Banach, M., 2015. Effect of sour tea (Hibiscus sabdariffa L.) on arterial hypertension. Journal of Hypertension, 33(6), pp.1119-1127

Wahabi, H., Alansary, L., Al-Sabban, A. and Glasziuo, P., 2010. The effectiveness of Hibiscus sabdariffa in the treatment of hypertension: A systematic review. Phytomedicine, 17(2), pp.83-86.

Round 2

Reviewer 1 Report

No more comment.

Author Response

(The authors gave the same response as above.)

Reviewer 3 Report

The logistics and circumstances to conduct such a study in Iraq in understood, and the limitations are noted. However, the bit about recommending the use of the herb should probably be removed in light of the several methodological limitations.